# TO UNDERSTAND REPRESENTATION OF LAYER-AWARE SEQUENCE ENCODERS AS MULTI-ORDER-GRAPH

## ABSTRACT

In this paper, we propose a unified explanation of representation for layer-aware neural sequence encoders, which regards the representation as a revisited multigraph called multi-order-graph (MoG), so that model encoding can be viewed as a processing to capture all subgraphs in MoG. The relationship reflected by Multi-order-graph, called $n$-order dependency, can present what existing simple directed graph explanation cannot present. Our proposed MoG explanation allows to precisely observe every step of the generation of representation, put diverse relationship such as syntax into a unifiedly depicted framework. Based on the proposed MoG explanation, we further propose a graph-based self-attention network empowered Graph-Transformer by enhancing the ability of capturing subgraph information over the current models. Graph-Transformer accommodates different subgraphs into different groups, which allows model to focus on salient subgraphs. Result of experiments on neural machine translation tasks show that the MoG-inspired model can yield effective performance improvement.

## 1 INTRODUCTION

Vaswani et al. (2017) propose self-attention (SAN)-based neural network (called Transformer) for neural machine translation (NMT). As state-of-the-art NMT model, several variants of the Transformer have been proposed for further performance improvement (Shaw et al., 2018; He et al., 2018) and for other natural language process tasks such as language model (Devlin et al., 2019), parsing (Kitaev & Klein, 2018; Zhou & Zhao, 2019), etc.

Similar as recurrent neural network (RNN)-based (Kalchbrenner & Blunsom, 2013; Bahdanau et al., 2015; Sutskever et al., 2014) model, SAN-based models try to make representation of one word containing information of the rest sentence in every layer. Empirically, one layer alone cannot result in satisfactory result, in the meantime, staking layers may greatly increase the complexity of model (Hao et al., 2019; Yang et al., 2019; Guo et al., 2019).

Better understanding the representations may help better solve the problem and further improve performance of SAN-based models. It is common to model the representation as a simple directed graph, which views words as nodes and relationships between words as edges. However, such understanding of representations may be still insufficient to model various and complicated relationship among words such as syntax and semantics, let alone presenting a unified explanation for the representations given by SAN- or RNN-based models (Eriguchi et al., 2016; Aharoni & Goldberg, 2017; Wang et al., 2018b). In addition, simple directed graph mostly models the relationship among words but is incapable of modeling the relationship among phrases or clauses.

To overcome the shortcomings of modeling the representation as a simple directed graph and then in the hope of helping further improve SAN-based model, in this paper, we propose a novel explanation that representation generated by SAN-based model can be viewed as a multigraph called *multi-order-graph* (MoG). In MoG, a set of nodes and edges between these nodes form a subgraph. Meanwhile, one edge not only connects words, but also connects subgraphs which words belong to. Thus we call the relationship reflected by MoG *n-order dependency*, where $n$ is the number of words involved in this relationship. With such an explanation, we can precisely observe every

step of the generation of representation, unify various complicated relationship such as syntax into $n$-order dependency and understand the model encoding eventually.

Inspired by our proposed explanation, we further propose a graph-based SAN empowered Graph-Transformer by enhancing the ability of capturing subgraph information over the current SAN-based sequence encoder. First of all, we generally define a **full representation** as the fusing result of all concerned subgraph representations. Then let the representation of one layer split into two parts, **previous representation** and **incremental representation**. The previous representation reflects full representation from previous layer, and the incremental representation reflects new information generated in this layer. Based on this, the encoding process is modified to adapt to such representation division. We split the original self-attention into three independent parts to generate incremental representation. Our method accommodates subgraphs of different orders into different parts of incremental representation, and reduces the information redundancy. To fuse the full representation, We consider three fusing strategies in terms of different weighting schemes so that let the model focus on salient parts of the representation.

## 2 MULTI-ORDER-GRAPH EXPLANATION

In graph theory, a directed multigraph (or pseudograph) is a graph has multiple parallel edges, and these edges have the same end nodes. Two vertices may be connected by more than one directed edge. In fact, multigraph is enough to reflect representation generated by model after encoding, while definition of edge cannot reflect relationship between subgraphs and the process of generation. In this paper, we propose a multigraph called multi-order-graph (MoG) for representation of input, which defines edges to reflect relationship between nodes more accurately.

### 2.1 ENCODING OF MODELS

General speaking, encoding of sentence is a process to transfer a sequence of words to a sequence of vectors. During encoding, model is treated as a stable function independent of data without change of parameters. Representation generated by model only reflects information of input sentence.

### 2.2 MULTI-ORDER-GRAPH

We generally define MoG as $G = (V, E, SN, TN)$ over a given sentence $S = \{s_1, ..., s_n\}$, in which nodes $V = \{v_1, ..., v_n\}$ reflect words of $S$, edges $E = \{e_1, ..., e_m\}$ reflect relationship between words of $S$, $SN = \{sn_1, ..., sn_m | sn_j \in V, 1 < j \leq m\}$ is the set of source node of each edge in $E$ and $TN = \{tn_1, ..., tn_m | tn_j \in V, 1 < j \leq m\}$ is the set of target node of each edge in $E$. Node $v_i \in V$ in $G$ can access other nodes in one step. Information captured from $S$ is splited into two parts, (1) **Word information**, which are contained in $V$ and reflects word, (2) **Relationship information**, which are contained in $E$ and reflect relationship of word-pairs.

Note that $E$ in $G$ is the most difference between MoG and standard multigraph. As mentioned above, MoG revises the definition of edges to reflect relationship between subgraphs of $G$. In Section 2.4 we will discuss the definition of edge $e_j \in E$, subgraph and relationship between edges and subgraphs in detail.

### 2.3 NODE AND WORD

Similar as simple directed graph, nodes in MoG reflect word of input sentence, which means number of nodes in MoG is equal to the number of words of input. Words are represented by nodes of MoG. Without relationship between words, MoG is just a set of graphs which have only one node and no edge. Obviously, one word is independent of others, and model cannot enrich word information.

### 2.4 EDGE AND SUBGRAPH

In this section, we define edge, subgraph and relationship between edge and subgraph in MoG.

A subgraph of $G$ is a graph whose vertex set is a subset of $V$, and whose edge set is a subset of $E$. We define $Sub_G = \{sub_1^G, ..., sub_p^G\}$ as the set of all subgraphs of $G$. Subgraph can be defined as

$sub_j^G = (V_j^G, E_j^G, SN_j^G, TN_j^G)$. Order of subgraph $sub_j^G$, which is equal to $|V_j^G|$, i.e., the number of nodes in it, and also means number of the words involved in this subgraph. The simplest subgraph has one node and no edge, and order of it is 1.

Edge in MoG connects two nodes which is same as simple direct graph. However, edges in MoG reflect not only relationship between words, but also the relationship between subgraphs. Given one node-pair, several edges are generated because nodes may belong to different subgraph-pairs. $p(v_i, v_i \in V_k^G | v_j, v_j \in V_h^G)$ is the conditional probability to present one relationship between $v_i$ and $v_j$. It indicates that edge $e_j$ is determined by four variables, (1) source node $sn_j$ of edge $e_j$, (2) target node $tn_j$ of edge $e_j$, (3) subgraph $sub_k^G$ in which $sn_j \in V_k^G$, (4) subgraph $sub_h^G$ in which $tn_j \in V_h^G$.

When $e_j$ is generated, $e_j$ will connect $sub_k^G$ and $sub_h^G$ and generate a novel subgraph, which we call this subgraph **related subgraph** of $e_j$ and use $sub_{R(j)}^G$ to represent it, where $R(j)$ is a function to get the identifier of related subgraph of $e_j$. To reflect importance of $e_j$ and complexity of $sub_{R(j)}^G$, we define order of $e_j$, which is represented by $o_j$ and equal to the order of $sub_{R(j)}^G$. We can use a 6-tuple

$$e_j = (sn_j, tn_j, sub_k^G, sub_h^G, sub_{R(j)}^G, o_j)$$

to present edge $e_j$. If we only focus on source and target node, we can use $(sn_j \rightarrow tn_j, o_j)$ for $e_j$.

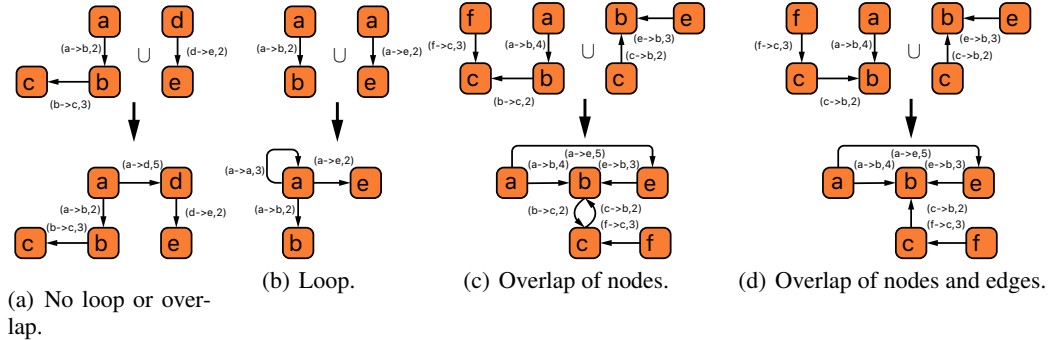

(a) No loop or overlap.

(b) Loop.

(c) Overlap of nodes.

(d) Overlap of nodes and edges.

Figure 1: Generation of different kinds of subgraph.

Figure 1 shows generation of four kinds of subgraph. To make the process of generating subgraph clear to understand, we only focus on subgraph without loop and overlap, which is the most simple kind of subgraph. Obviously, subgraphs and edges cannot be generated in random order. The order in which edges are generated is the order in which related subgraph of these edges are generated, which is also the order in which all subgraphs are generated. It also means that the process of subgraph (edge) generation is an iterative process, in which one subgraph (edge) relies on previous generated subgraphs (edges).

We define an operation to express the process of generating subgraphs,
$$sub_k = (sub_i) \rightarrow v_m \cup (sub_j) \rightarrow v_n,$$

This operation means that one new edge $(v_m, v_n, sub_i, sub_j, sub_k, |V_i| + |V_j|)$ and one new subgraph $sub_k$ are generated, where $|V_i|$ and $|V_j|$ are orders of $sub_i$ and $sub_j$, $v_m \in sub_i$ is the source node of new edge, $v_n \in sub_j$ is the target node of new edge and $sub_k$ is generated by connecting $sub_i$ and $sub_j$. Note that the commutative property, distributive property and associative property do not apply in this formula. For an example, the process of generating subgraph in Figure 1(a) can be expressed as
$$(((sub_a) \rightarrow v_a \cup (sub_b) \rightarrow v_b) \rightarrow v_b \cup (sub_c) \rightarrow v_c) \rightarrow v_a \cup ((sub_d) \rightarrow v_d \cup (sub_e) \rightarrow v_e) \rightarrow v_d,$$

where $sub_a$, $sub_b$, $sub_c$, $sub_d$ and $sub_e$ are subgraphs with only one node. It also means that this process can be expressed as binary tree. Especially, given a sentence, if we add words to one subgraph according to the order of word in the sentence, this subgraph can reflect the order of the sentence.

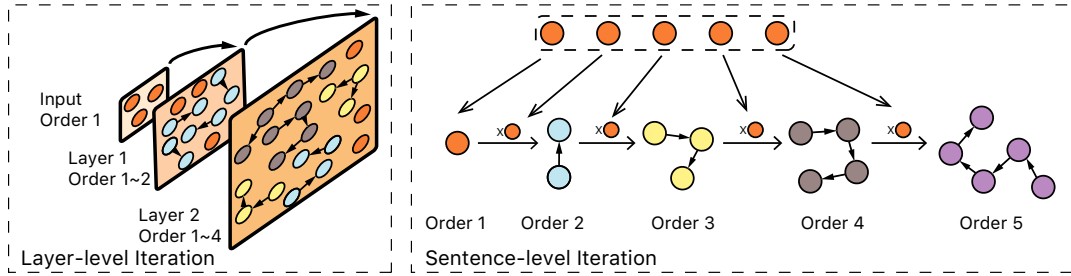

Figure 2: Iteration.

As mentioned above, relationship which $e_j$ reflects is not only relationship between words but also relaitonship between subgraphs. In this paper, we call this relationship, which is a combination of relationship between words and between subgraphs, $n$-**order dependency** where $n$ is equal to $o_j$. In fact, $n$-order dependency relationship can conveniently model quite a lot of relationships among words, typically, various syntax.

### 2.5  FOUR KEY QUESTIONS FOR MODELS

Based on the proposed MoG explanation, an effective model should better generate edges and capture subgraphs iteratively, which triggers the following four basic questions.

• **How to preserve subgraph information?** Neural models have to use vector for this purpose, which lets the dimension of vectors become an important factor to define the vector accommodation.

• **How to implement iterative encoding?** Two kinds of iteration are shown in Figure 2. Sentence-level iteration allows model to encode words one by one as used in RNN-based model. With sentence-level iteration, the order of subgraph is the sentence length. All layer-aware models implement layer iteration by generating representations in one layer to feed next layer.

• **How to capture edges and subgraphs?** RNN-based models use recurrent networks, convolutional neural network based models (Gehring et al., 2017; Dauphin et al., 2017) use convoluation+gating blocks, and SAN-based models use self-attention.

• **Which architecture will be selected?** A proper architecture can focus on advantages of solutions of other three questions.

These four questions are correlated to each other. Using vector causes dimension of vector important. Iteration makes number of layer important and affects architecture of model. Methods to generate edges is related to a proper iteration and architecture.

Inspired by the MoG explanation for the representations and following the path well answering the related four questions, we may further improve the encoder design. Thus we propose a graph-aware Transformer as follows.

## 3  GRAPH-TRANSFORMER

### 3.1  MULTI-ORDER-GRAPH IN THE TRANSFORMER

SAN-based models use self-attention to capture edges and subgraphs, and use layer-level iteration only. Regarding representation as a MoG $G$, we can use a one-dimensional matrix to contain all information of $G$, which means that we may use a matrix with the same shape as representation to represent a subgraph. Given representation for a set of subgraphs, one representation-pair can be expanded into a set of subgraph-pair. Given representations $r_a$ and $r_b$, $\{sub_1^a, ...sub_n^a\}$ are $n$ subgraphs contained in $r_a$ and $\{sub_1^b, ...sub_m^b\}$ are $m$ subgraphs contained in $r_b$

$$r_a = \sum_i^n sub_i^a, r_b = \sum_i^m sub_i^b.$$

Self-attention has to get an attention matrix $\mathcal{M}$ using query and key accoring to Equation (3). In the $i$-th layer of SAN-based model, representation of word $s_m$ generated by this layer is $r_m^i$ and

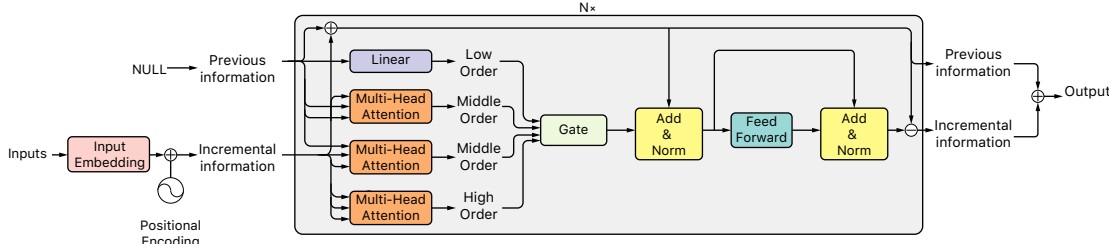

Figure 3: Graph-Transformer.

attention matrix is $\mathcal{M}_i$. To calculate matrix $\mathcal{M}_i$, if $r_a$ is query and $r_b$ is key,

$$r_a \cdot (r_b)^\top = (\sum_i^n sub_i^a) \cdot (\sum_i^m sub_i^b)^\top = \sum_i^n \sum_j^m sub_i^a \cdot (sub_j^b)^\top \tag{1}$$

Value in $\mathcal{M}_i$ reflects relationship between two words which can be reflect as edge in MoG. Equation (1) shows that relationship of all subgraph-pairs which are extended from $r_a$ and $r_b$ can be calculated at once and new subgraphs by connecting old subgraphs from $r_a$ and $r_b$ are also generated.

In the $i$-th layer, representations used as query, key and value are from the $i-1$-th layer, which means that subgraphs generated by $i-1$-th layer will affect the largest order of subgraphs in $i$-th layer. Connecting two input subgraphs of the largest order will generate a subgraph of the largest order in $i$-th layer, which makes the largest order of subgraphs increases exponentially as layers increases and the largest order of subgraphs in $i$-th layer is $2^i$.

However, SAN-based model cannot accurately capture all subgraphs and obtain information of input because the largest order of subgraphs is limited by the number of layers. Information obtained by $n$-layer model cannot be complete if the length of input exceeds $2^n$.

Naturally, models always capture subgraphs of low order repeatedly. A generated subgraph is always contained in representation and used to generate new subgraphs. As a result, the earlier subgraph is generated, the more times it will be generated. It increases weight of subgraphs of low-order in a latent way. Besides, saving multiple information of subgraphs in one vectors makes model difficult to distinguish them and extract salient subgraphs from vectors.

### 3.2 Architecture of Graph-Transformer

Figure 3 is the architecture of our graph-Transformer. First, we define a **full representation** for the fusing results of all concerned subgraph information. We split full representation generated by one layer into **previous representation** and **incremental representation** to group subgraphs. Previous representatin is full representation from previous layer which is also the sum of input previous and incremental representations. Incremental representation is designed for new information generated by one layer. To generate incremental representation, we split self-attention into three parts to generate subgraphs of different order. A gate is used to adjust weights of different groups of subgraphs. The sum of previous and incremental representations is the final representation of one layer.

### 3.3 Self-Attention Group for Subgraphs of Different Order

The original SAN-based model uses input representation as query, key and value to calculate self-attention. Split input representation into previous representation and incremental representation, it is obvious that calculation of self-attention can be viewed as the sum of four parts

$$r_f \cdot r_f^\top = (r_p + r_i) \cdot (r_p + r_i)^\top = r_p \cdot r_p^\top + r_p \cdot r_i^\top + r_i \cdot r_p^\top + r_i \cdot r_i^\top$$

where $r_f$ is full representation, $r_p$ is previous representation and $r_i$ is incremental representation. Note that $r_p$ is also the $r_f$ of the previous layer, which means that $r_p \cdot r_p^\top$ has been calculated by previous layer and makes subgraphs be generated repeatedly. It is also the key to increase the weight of subgraphs of low-order. To avoid redundancy, we only calculate other three parts of self-attention. There are three levels for the subgraph order.

• **High order.** Subgraphs generated by $r_i \cdot r_i^\top$ belong to high order, and one part of self-attention is used to process subgraph of high order, which uses input incremental representation as its query, key and value. In the $i$-th layer, the order of subgraph is in the range of $2^{i-1}$ to $2^i$.

• **Middle order.** Subgraphs generated by $r_p \cdot r_i^\top$ and $r_i \cdot r_p^\top$ belong to middle order and other two parts of self-attention. The second part of self-attention uses input incremental representation as query and input previous representation as key and value. The third part of self-attention uses input previous representation as query and input incremental representation as key and value. In the $i$-th layer, the order of subgraph is in the range of $2^{i-2}$ to $2^{i-1}$.

• **Low order.** Subgraphs generated by $r_p \cdot r_p^\top$ belong to low order. As we discuss above, it is no need to calculate $r_p \cdot r_p^\top$ again. Instead of self-attention, we use a linear function for transformation of vector space. The subgraph order is in the range of 1 to $2^{i-2}$

To reduce number of parameters and avoid overfitting, we share vector of query, key and value in three parts of self-attention, while it is difficult to train such a model because different group of subgraphs requires different vector spaces. We can also drop the dimension of model in self-attention. To keep the least effect over the performance, we reduce dimension of model to half of original dimension.

## 3.4 Fusion of Representations

To get the full representation, we introduce three fusing strategies to combine previous and incremental representation. Calculating the sum is the most simple strategy. However, this strategy depends on the quality of previous representation and incremental representation. Besides, model gives four groups of subgraphs equal weights, which cannot weight important subgraphs.

Representations generated by self-attention are new subgraphs which have not been weight by model. Viewing these three parts of representation as one group, we can use a gate to calculate their importance and merge them.
$$w = \text{Sigmoid}(i_h + i_m + i_l), r_f = (i_h + i_m) \cdot w + i_l \cdot (1 - w),$$

where $i_h$, $i_m$ and $i_l$ are subgraph of high order, middle order and low order. With gate to assign weight, model can explicitly distinguish new and old subgraph and pay attention on important group of subgraph. Disadvantage of this method is that the model cannot distinguish subgraph of high and middle order. We call this method **weight-gate**.

Wang et al. (2018a) propose a fusion function based on self-attention with hops for fusion of representation of different layers. Similar as Wang et al. (2018a), we use self-attention to generate a matrix of weight which stands for relationship between representations. To assign weight of four representations, we concatenate four representations to form a new sequence $R$ and calculate the matrix of relationship between different representation.
$$R_f = softmax(R_q R_k^T / d_k) R_v / 4,$$

where $R_f$ is the representation sequence, $R_q$, $R_k$ and $R_v$ are vector of query, key and value, $d_k$ is the dimension of model. This method can capture relationships between representations and weight them. Weight of one group will be larger if it is more important than others. To make sum of weight equal to 1, representation is divided by 4. We call this method **self-gate**. Self-gate can weight all representations by model while according to the function of self-attention, self-gate will generate subgraphs of higher order which makes model deeper and more difficult to be trained.

## 4 Experiments and Results

### 4.1 Neural Machine Translation

In this paper, we evaluate our model on four machine translation tasks, IWSLT14 German-English (De-En), WMT14 English-German (En-De), WMT14 English-French (En-Fr) and WMT16 English-Romanian(En-Ro).

Our baselines for En-De, En-Fr and En-Ro are Transformer-base, and baseline for De-En is Transformer-small. Without the proposed graph mechanism, our model is a variant of the Trans-

| Model | De-En | En-De | En-Fr | En-Ro |
|---|---|---|---|---|
| | BLEU | BLEU | BLEU | BLEU |
| Existing NMT systems | | | | |
| Transformer (small) (He et al. (2018)) | 32.9 | - | - | - |
| Transformer (base) (He et al. (2018)) | - | 27.3 | - | - |
| Transformer (base) (Shaw et al. (2018)) | - | 26.5 | 38.2 | - |
| He et al. (2018) | 35.1 (+2.2) | 28.3 (+1.0) | - | - |
| Shaw et al. (2018) | - | 26.8 (+0.3) | 38.7 (+0.5) | - |
| Our NMT systems | | | | |
| Transformer(small) | 36.5 | - | - | - |
| Transformer(base) | - | 27.1 | 40.1 | 33.9 |
| Graph-Transformer | 37.1 (+0.6) | 27.5 (+0.4) | - | - |
| +half-dim & gate | 37.2 (+0.7) | 28.2 (+1.1) | 40.8 (+0.7) | 34.6 (+0.7) |

Table 1: Multi-BLEU scores on De-En, En-De, En-Fr and En-Ro. The baselines are Transformer-small and Transformer-base, respectively.

| Model | De-En | | | En-De | | | |
|---|---|---|---|---|---|---|---|
| | BLEU | #Para | #Speed | BLEU | #Para | #Speed | #PPL |
| Transformer(small) | 36.5 | 42M | 50K | - | - | - | - |
| Transformer(base) | - | - | - | 27.1 | 66M | 137K | 4.92 |
| Graph-Transformer | 37.1 | 50M | 42K | 27.5 | 77M | 112K | 4.92 |
| +half-dim | 37.5 | 47M | 39K | 27.4 | 71M | 107K | 4.95 |
| +gate | 37.3 | 57M | 39K | 28.0 | 80M | 109K | 4.94 |
| +self-gate | 36.9 | 53M | 30K | 27.6 | 77M | 91K | - |
| +shared-qkv & gate | 37.1 | 51M | 40K | 27.7 | 75M | 121K | 4.91 |
| +half-dim & gate | 37.2 | 50M | 35K | 28.2 | 74M | 111K | 4.89 |
| +half-dim &gate &shared-qkv | 37.5 | 47M | 38K | 27.7 | 70M | 115K | 4.93 |

Table 2: Multi-BLEU scores of ablations on De-En and En-De. #Para, #Speed, #Mem and #PPL denote the size of model paragraphs, training speed (tokens/second), GPU memory model used (GB) and perplexity respectively.

former with three parts of self-attention. We test several methods such as half-dimension (half-dim), weight-gate (gate), shared-query-key-value (shared-qkv) and self-gate (self-gate). We further compare our model with Shaw et al. (2018) and He et al. (2018).

• **Shaw et al. (2018)** introduce relative position encoding which adds the relative distance into the representation.

• **He et al. (2018)** propose a method to coordinate the learning of hidden representations of the encoder and decoder together layer by layer.

Table 1 compares our Graph-Transformer with the baseline, showing that our model enhances both tasks and outperforms all baselines. For De-En tasks, our model with half-dimension gets the best performance of 37.5 BLEU points with 47 million parameters. For En-De tasks, our model with half-dimension and weight-gate gets the best performance of 28.2 BLEU points outperforming the Transformer-base by 1.1 BLEU points with 74 million paramenters. For En-Fr and En-Ro, our model with half-dimension and weight-gate gets hte performance of 40.8 BLEU points outperforming the Transformer-base by 0.7 BLEU points and 34.63 BLEU points outperforming the Transformer-base by 0.7 BLEU points respectively. With a baseline of 27.1 BLEU point on En-De and 40.1 BLEU point on En-Fr, the improvement of Graph-Transformer is better than Shaw et al. (2018) and He et al. (2018) on En-De and En-Fr tasks.

Fusion methods perform differently on De-En and En-De. For De-En tasks, self-gate gets the lowest performance of 36.9 BLEU points. For En-De tasks, calculating the sum gets the lowest performance

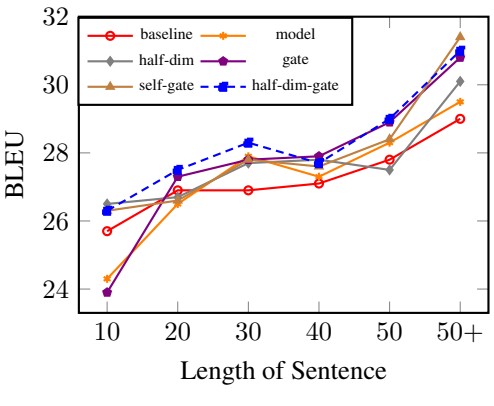
(a) BLEU points of different lengths.

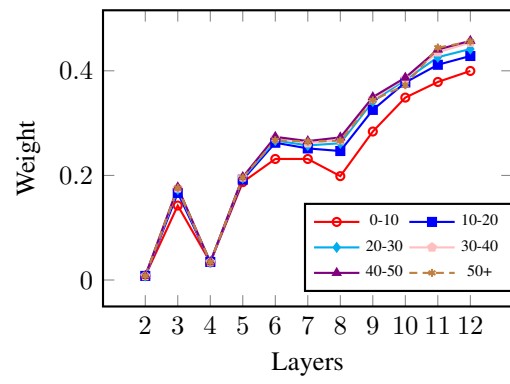
(b) Subgraph weights vs. sentence lengths in different layers.

Figure 4: The effect of sentence length and layers on performance

of 27.5. For both tasks, weight-gate is the most effective among all fusing methods. Using weight-gate to weight different groups of subgraphs has shown indeed helpful.

Figure 4(a) shows the relationship between performance and length of input sentences. With the longer input sentences, all our methods outperform baseline.

Figure 4(b) shows that how weight of subgraph of high order change in different layers with different length of sentence. This result is based on one 12-layer model with combination of half-dimension and gate. The trend of weight change means that our model will pay more attention on subgraphs of higher order in higher layers of model. In the appendix, Figure 5(h) means that the trend of weight change is independent of number of layers.

## 4.2 NAMED-ENTITY RECOGNITION, PART-OF-SPEECH TAGGING, TEXT SUMMARIZATION

We also conduct new experiments on Named-entity recognition (NER), part-of-speech tagging (POS tagging) and text summarization. We use CoNLL-2003 Sang & Meulder (2003) dataset, Wall Street Journal (WSJ) Corpus and Annotated Gigaword Napoles et al. (2012) corpus as benchmark of NER task, POS tagging task and text summarization respectively. In these experiments, we use the Transformer as the baseline and evaluate our model with halfdim-gate. Table 3 shows the result of NER and POS tagging. Table 4 shows the result of text summarization.

|  | NER (F1) | POS tagging (F1) |
|---|---|---|
| Transformer | 80.2 | 96.40 |
| Graph-Transformer (halfdim&gate) | 80.3 (+0.1) | 96.45 (+0.05) |

Table 3: Results of NER and POS tagging

|  | ROUGE-1 | ROUGE-2 | ROUGE-L |
|---|---|---|---|
| Transformer (base) | 36.84 | 18.01 | 34.31 |
| Graph-Transformer (halfdim&gate) | 37.38 (+0.54) | 18.59 (+0.58) | 34.58 (+0.27) |

Table 4: Results of text summarization.

For NER and POS tagging tasks, Graph-Transformer improves the performance tiny. For text summarization, our model with halfdim-gate outperforms the baseline on all evaluation metrics.

In NER and POS tagging tasks, lengths of sentences in the dataset is often short and make the advantages of Graph-Transformer useless. As we discussed in Section 3.1, the largest order of subgraphs generated by model is limited by the number of layer while it is also limited by the length of sentence. Subgraphs of high order will be generated early and weight of these subgraphs will be increased in following layer.

## 5 ANALYSIS OF RESULT

According to MoG explanation and the design of Graph-Transformer, not calculating subgraph of low order can avoid generating subgraph repeatedly. It ensures that every subgraph to only be generated once and has the same weight, so that the model performance should be slightly improved. Meanwhile, weighting subgraphs allows model to figure out salient subgraphs. Without weighting subgraph, our model can only outperform baseline 0.4 BLEU points in En-De task, and outperform baseline more than 0.9 BLEU points after weighting subgraph by weight-gate, which is the same as we expected and indicates the reasonableness of our MoG explanation.

**Fusion Methods.** Table 1 compares different fusion methods, in which weight-gate gets the best performance, while self-gate is not the best one as expected. Calculating sum of representation only makes every subgraph be generated once and have the same weight while it cannot allow model to figure out salient subgraphs. This method also performs worst.

Self-gate can weight every group of subgraphs which cannot be done by weight-gate. However, using self-attention and representation, self-gate will generate new subgraphs of high order and unnecessary redundancy. Self-gate also makes model deeper and difficult to train.

Although weight-gate cannot distinguish every subgraph in representation, it makes model focus on specific parts. In fact, using the same query, key and value to produce representations, there are some stable relationship between them. Dividing them into two groups can mostly distinguish this relationship and allow model to capture it.

**Dimension of Models.** Table 1 shows that model with half dimension can get a similar or better result compared with model with full dimension. Larger model dimension enables vector to accommodate more features. Our results do not mean larger dimension is unimportant. Though we use less parameters, our model can capture subgraph more accurately. Our model can distinguish subgraph with different orders with three independent parts of self-attention.

Besides, more parameters usually let model more difficultly trained and more easily be overfitting. Thus half dimension setting helps the resulted model to outperform the model with full dimension.

## 6 RELATED WORK

Several variants have been proposed to improve performance of the original SAN-based model. Shaw et al. (2018) proposed relative position representations in the self-attention mechanism to replace the absolute position encoding and it enhances the ability of capturing local information of the input sentence. He et al. (2018) shared the parameters of each layer between the encoder and decoder to coordinate the learning between encoder and decoder. BERT (Devlin et al., 2019) is a language model which is to pre-train deep bidirectional representations from unlabeled text by jointly conditioning on both left and right context in all layers. Dai et al. (2019) enabled the Transformer to learn dependency beyond a fixed length without disrupting temporal coherence. Koncel-Kedziorski et al. (2019) propose a Graph-based model on text generation. Zhu et al. (2019) use graph structures for AMR. Cai & Lam (2020) propose a graph structure network for AMR.

## 7 CONCLUSIONS

This paper presents a unified explanation for representations given by sequence encoders, especially, the SAN empowered Transformer. Instead of a simple directed graph modeling in previous work, we re-define multigraph into multi-order-graph to accommodate a broad categories of complicated relationships inside the representations. MoG connects not only words but also subgraphs. With the built relationship by MoG, which is called $n$-order dependency, we can understand diverse relationships inside representations as complicated as syntax in a unified way. Inspired by the proposed MoG explanation, we further propose a graph-Transformer to enhance the ability of capturing subgraph information on the SAN-based encoder. Experimental results indicate that our proposed MoG explanation for representations is empirically reasonable.

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

## A APPENDIX

### A.1 BACKGROUND OF GRAPH-TRANSFORMER

Transformer (Vaswani et al., 2017) is state-of-the-art NMT model empowered by self-attention networks (SANs) (Lin et al., 2017), in which an encoder consists of one self-attention layer and a position-wise feed-forward layer, decoder contains one self-attention layer, one encoder-decoder attention layer, and one position-wise feed-forward layer. SAN-based model is also used in other tasks such as language model (Devlin et al., 2019), parsing (Kitaev & Klein, 2018; Zhou & Zhao, 2019), etc. SAN-based model uses residual connections around the sublayers followed by a layer normalization layer.

The encoder reads an input sentence, which is a word sequence $x = \{x_1, ...x_{T_x}\}$, and encodes it as a context vector $c$. Decoder is trained to predict the next word given the context vector generated by encoder and all previously predicted words $\{y_1, ..., y_{t-1}\}$. The decoder defines a probability over the translation $y$ by decomposing the joint probability into the ordered conditionals,

$$p(y) = \prod_{t=1}^{T_y} p(y_t \mid \{y_1, \cdots, y_{t-1}\}, c). \tag{2}$$

Scaled dot-product attention is the key component in Transformer. The input of attention contains queries ($Q$), keys ($K$), and values ($V$) of input sequences. The attention is generated using queries and keys like Equation (3),

$$Attention(Q, K, V) = \text{softmax}(\frac{QK^T}{\sqrt{d_k}})V. \tag{3}$$

Different from RNN-based models which process words/subwords one by one, dot-product attention allows Transformer to generate the representation in parallel.

Vaswani et al. (2017) also propose multi-head attention which generates representation of sentence by dividing queries, keys, and values to different heads and gets representative information from different subspaces.

Quality of encoding sentence and generating representation can influence performance of NMT model significantly. RNN-based and SAN-based models use different mechanisms to implement encoding, thus show different natures for the resulted representation. RNN-based model is good at capturing localness information and not good at parallelization and long-range dependency capturing, while SAN-based model is better at capturing long-range dependencies with excellent parallelization.

### A.2 DATASETS OF EXPERIMENTS

**IWSLT14 German-English** IWSLT14 De-En dataset contains 153K training sentence pairs. We use 7K data from the training set as validation set and use the combination of dev2010, dev2012, tst2010, tst2011 and tst2012 as test set with 7K sentences which are preprocessed by script[1]. BPE algorithm is used to process words into subwords, and number of subword tokens is 10K.

---

[1] https://github.com/pytorch/fairseq/blob/master/examples/translation/prepare-iwslt14.sh

| Parameter | DE-EN | EN-DE |
|-----------|-------|-------|
| Layers | 6 | 6 |
| Dimension | 512 | 512 |
| Head | 4 | 8 |
| FF | 1024 | 1024 |
| Dropout | 0.3 | 0.1 |

Table 5: Hyperparameters for our experiments. FF is short for feed-forward layer.

**WMT14 English-German** We use the WMT14 En-De dataset with 4.5M sentence pairs for training. We use the combination of newstest2012 and newstest2013 as validation set and newstest2014 as test set which are preprocessed by script[2]. The sentences longer than 250 are removed from the training dataset. Dataset is segmented by BPE so that number of subwords in the shared vocabulary is 40K.

### A.3 HYPERPARAMETERS OF EXPERIMENT

The hyperparameters for our experiments are shown in Table 5. For De-En, we follow the setting of Transformer-small. For En-De, we follow the setting of Transformer-base.

### A.4 TRAINING OF EXPERIMENT

Our models for En-De, En-Fr and En-Ro are trained on one CPU (Intel i7-5960X) and four nVidia RTX TITAN X GPUs, and models for De-En are trained on one CPU (Intel i7-5960X) and one nVidia RTX TITAN X GPU. The implementation of model for NMT tasks is based on fairseq-0.6.2. We choose Adam optimizer with $\beta_1 = 0.9$, $\beta_2 = 0.98$, $\epsilon = 10^{-9}$ and the learning rate setting strategy, which are all the same as Vaswani et al. (2017),

$$lr = d^{-0.5} \cdot \min(step^{-0.5}, step \cdot warmup_{step}^{-1.5}),$$

where $d$ is the dimension of embeddings, $step$ is the step number of training and $warmup_{step}$ is the step number of warmup. When the number of step is smaller than the step of warmup, the learning rate increases linearly and then decreases.

We use beam search decoder for De-En task with beam width 6. For En-De, following Vaswani et al. (2017), the width for beam search is 6 and the length penalty $\alpha$ is 0.2. The batch size is 1024 for De-En and 4096 for En-De. We evaluate the translation results by using multiBLEU.

### A.5 AN EXAMPLE OF MODELING SYNTAX IN MoG

Figure 7 shows an example of modeling syntax in MoG. Similar as this example, one syntactic tree can be viewed as one subgraph of MoG. Generating a syntactic tree can be viewed as a process of generating a subgraph. We can use an equation to express the example,

$$
\begin{aligned}
sub_{sens} =& (((sub_{Do}) \to v_{Do} \cup (sub_{do}) \to v_{do}) \to v_{do} \\
& \cup ((sub_{Romans}) \to v_{Romans} \cup (sub_{the}) \to v_{the}) \to v_{Romans}) \to v_{do} \\
& \cup (sub_{as}) \to v_{as}
\end{aligned}
\tag{4}
$$

where $sub_{sens}$ is the subgraph to reflect whole sentence.

### A.6 LAYER-LEVEL AND SENTENCE-LEVEL ITERATION

As we discussed in Section 2.4, process of subgraph (edge) generation is an iterative process, in which one subgraph (edge) relies on previous generated subgraphs (edges). Here is one question for this operation,

- **Where did previous generated subgraphs come from?**

---

[2]https://github.com/pytorch/fairseq/blob/master/examples/translation/prepare-wmt14en2de.sh

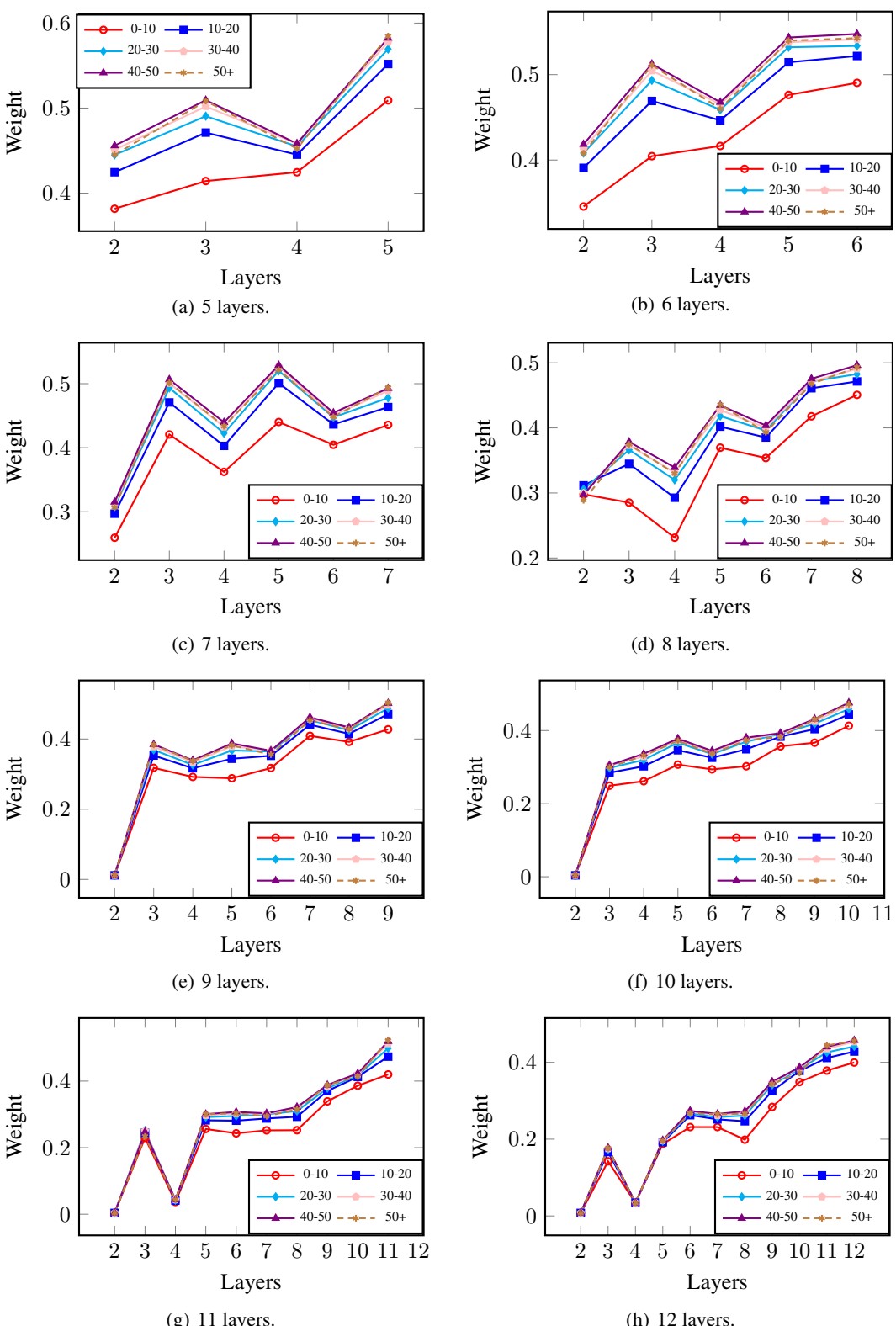

Figure 5: Subgraph weights in models with different layers on WMT14 En-De.

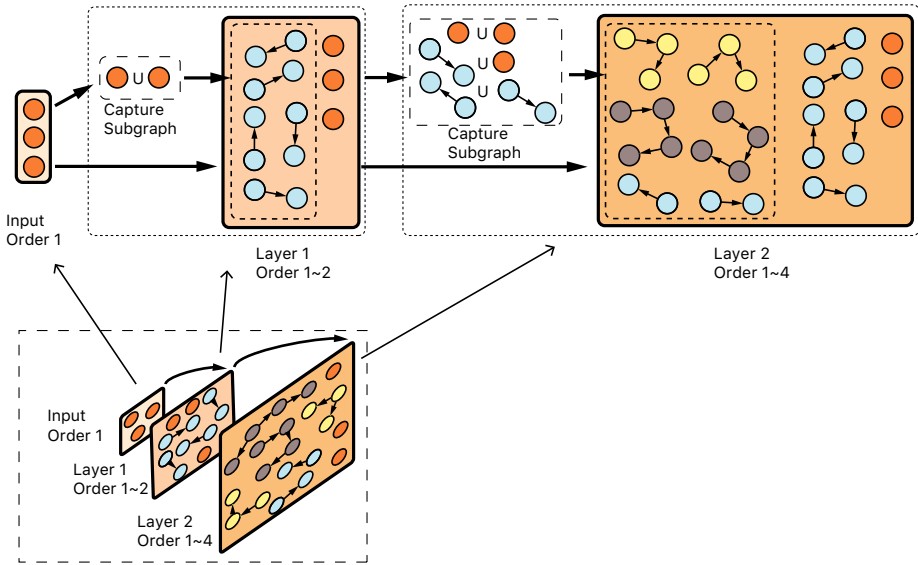

Figure 6: An example of layer-level iteration in Figure 2. Note that in one layer, layer-level iteration only operates once which only takes output of previous layer as input and generates new subgraphs.

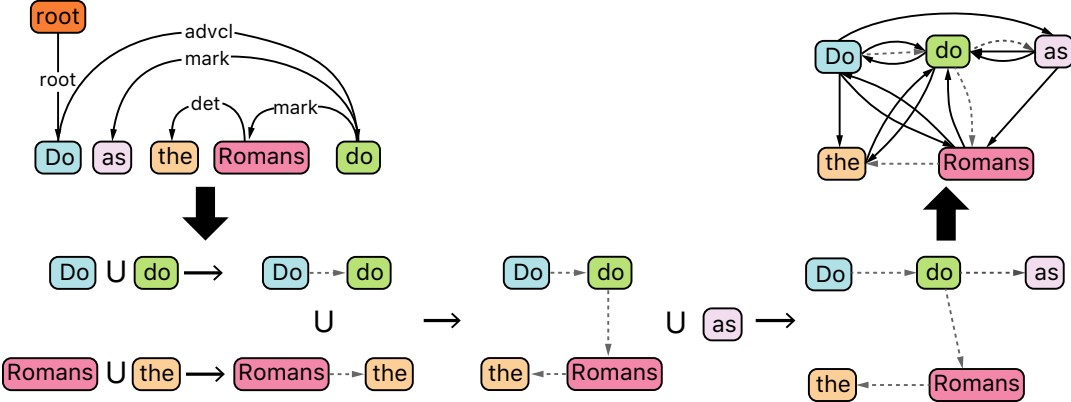

Figure 7: An example of modeling syntax in MoG.

which can also be transferred to another question,

- **Where will subgraphs generated be used to capture new subgraphs?**

Two kinds of iteration are answers to the question. Layer-level iteration means that one layer uses subgraph generated by previous layer to capture subgraph, and generated subgraph will be used in next layer. Sentence-level iteration means that one layer uses subgraphs generated in the same layer at previous time step, and generated subgraph will be used in the same layer at next time step.

Note that layer-level iteration is not the opposite of sentence-level iteration. They can be used together in one model, or can exist independently in one model. However, in most of layer-aware model with multiple layers, layer-level iteration is necessary.

### A.7 MoG Explanation for RNN

MoG can also explain encoding of RNN-based model, and analysing RNN-based model with four questions in Section 2.5 is a good start. Considering the various variants of RNN, we focus on GRU-based model and give an explanation of GRU-based model as an example.

- **How to preserve subgraph information?** GRU-based model uses vector to preserve subgraph information same as other neural models.

- **How to implement iterative encoding?** GRU-based model uses Layer-level iteration and Sentence-level iteration, which means that (1) GRU-based model can capture order information directly from the encoding. (2) one layer is enough to capture subgraphs of largest order theoretically.

- **How to capture edges and subgraphs?** GRU-based model uses one GRU cell to capture edges and subgraphs. The key component in GRU cell to capture edges is gate.

- **Which architecture will be selected?** Architecture of GRU-based model is simple which is only a stack of GRU layer.

Given input sentence $(x_1, ..., x_t)$, one layer of GRU-based model encodes words sequentially and representation of one word depends on representation of previous words

$$
\begin{aligned}
h_t^i &= f(h_t^{i-1}, h_{t-1}^i), \\
c^i &= q(\{h_1^i, ..., h_t^i\})
\end{aligned}
\tag{5}
$$

where $h_t^i \in^n$ is a hidden state of $i$-th layer at time step $t$, and $c^i$ is a vector generated from the sequence of the hidden states of $i$-th layer. $f$ and $q$ are some nonlinear functions. Time $t$ is also position of words in the sentence.

The GRU transition equations are the following

$$
\begin{aligned}
z_t^i &= \sigma(W_i^{(z)} h_t^{i-1} + U_i^{(z)} h_{t-1}^i + b_i^{(z)}), \\
r_t^i &= \sigma(W_i^{(r)} h_t^{i-1} + U_i^{(r)} h_{t-1}^i + b_i^{(r)}), \\
\widetilde{h}_t^i &= tanh(W_i^{(\widetilde{h})} h_t^{i-1} + U_i^{(\widetilde{h})} h_{t-1}^i + b_i^{(\widetilde{h})}), \\
h_t^i &= (1 - z_t^i) \odot h_{t-1}^i + z_t^i \odot h_t^{i-1}
\end{aligned}
\tag{6}
$$

where $z_t^i$ and $r_t^i$ is the update gate and reset gate of $t$-th time step in $i$-th layer respectively, and $h_t^i$ is the hidden state of $t$-th time step in $i$-th layer.

Besides, the LSTM transition equation are the following

$$
\begin{aligned}
in_t^i &= \sigma(W_i^{(in)} h_t^{i-1} + U_i^{(in)} h_{t-1}^i + b_i^{(in)}), \\
f_t^i &= \sigma(W_i^{(f)} h_t^{i-1} + U_i^{(f)} h_{t-1}^i + b_i^{(f)}), \\
o_t^i &= \sigma(W_i^{(o)} h_t^{i-1} + U_i^{(o)} h_{t-1}^i + b_i^{(o)}), \\
u_t^i &= tanh(W_i^{(u)} h_t^{i-1} + U_i^{(u)} h_{t-1}^i + b_i^{(u)}), \\
c_t^i &= in_t^i \odot u_t^i + f_t^i \odot c_{t-1}^i, \\
h_t^i &= o_t^i \odot tanh(c_t^i)
\end{aligned}
\tag{7}
$$

where $in_t^i$, $f_t^i$ and $o_t^i$ are the input gate, forget gate and output gate of $t$-th time step in $i$-th layer respectively, $c_t^i$ is the state of cell and $h_t^i$ is the hidden state of $t$-th time step in $i$-th layer.

LSTM and GRU are depended on gate

$$
W_i h_t^{i-1} + U_i h_{t-1}^i + b_i
\tag{8}
$$

Given representation for a set of subgraphs, one representation-pair can be expanded into a set of subgraph-pair. Given representation $r_a$ and $r_b$, $\{sub_1^a, ...sub_n^a\}$ are $n$ subgraphs contained in $r_a$ and $\{sub_1^b, ...sub_m^b\}$ are $m$ subgraphs contained in $r_b$

$$
r_a = \sum_i^n sub_i^a, \quad r_b = \sum_i^m sub_i^b,
\tag{9}
$$

Equation of gate can be transferred to

$$
\begin{aligned}
Wr_a + Ur_b + b &= W \sum_{i}^{n} sub_i^a + U \sum_{j}^{m} sub_j^b + b \\
&= \frac{1}{m} \cdot m \cdot W \sum_{i}^{n} sub_i^a + \frac{1}{n} \cdot n \cdot U \sum_{j}^{m} sub_j^b + b \\
&= m \cdot W \sum_{i}^{n} \frac{sub_i^a}{m} + n \cdot \sum_{j}^{m} U \frac{sub_j^b}{n} + b \\
&= \sum_{i}^{n} \sum_{j}^{m} (W \frac{sub_i^a}{m} + U \frac{sub_j^b}{n}) + b
\end{aligned}
\tag{10}
$$

which means that calculating relationship between $r_a$ and $r_b$ can be transferred to calculating relationship all subgraph-pairs of $r_a$ and $r_b$. Relationship of $sub_i^a$ and $sub_j^b$ is calculated by gate. New representation generated by LSTM or GRU is also a set of subgraph information.

Note that weights of edges always decrease with the increase of subgraphs. It causes that the earlier subgraph is generated, the larger its weight is, which is similar as SAN-based model. Different from RNN-based model, weight of subgraph is reflected by the number of times it is repeatedly generated.

RNN-based model uses layer-level and sentence-level iteration. Different from SAN-based model which uses more layers to generate subgraphs of higher order, RNN-based model can generate subgraphs of largest order in one layer. Therefore, design of deeper model is more suitable for SAN-based model. Even if the SAN-based model cannot capture subgraph of higher order, it can also adjust and balance the weight of subgraph using more layers.

## A.8    USING MoG EXPLANATION TO SOLVE EXISTING PROBLEMS

### A.8.1    WHAT IS POSITION-ENCODING IN SAN-BASED MODEL?

Position-encoding is a component to mark the position of words. In SAN-based model, position encoding is implemented as one-dimensional vector which can also be viewed as a special feature, and is combined into the representation of input which becomes one part of **word information** discussed in Section 2.2. Encoding of input sentence can generate relationship between different positions while it cannot reflect order of input sentence directly. Relationship between positions is contained in representation, which is also contained in edge from the perspective of MoG.

Position and order of words are two related property while they are not exactly the same. In some cases, these two property cannot be converted to each other. Position is an absolutely fixed property which can exist independent from other positions, while order is an relatively property which must depend on other words. In NLP, position of words can be converted to order of words because the input sentence is usually a continuous uninterrupted one-dimensional word sequence and the value of position is often one integer which can obtain order by comparing the value of position. It is also the reason why SAN-based model try to use position-encoding to make use of the order of the sequence.

Although there are various methods to indicate the location information in position-encoding and it is easy for human to convert position to order, it is difficult for SAN-based model to effectively obtain the order of sequence from location information. We conduct an experiment on IWSLT 14 De-En to show that SAN-based model views position-encoding as a feature and cannot capture indirect order information of sequence which we put in position-encoding. In our experiment, we replace well organized and designed position-encoding with another position-encoding which is designed by us. Without any well organized functions to generate vector for position-encoding, we full all vector of position-encoding with random value in a range from 0 to 1, which means that there is no meaningful information in position-encoding. Table 6 shows the result of experiment that position-encoding with random value does not impact performance negatively.

|            | Transformer | Transformer (no PE) | Transformer (random PE) |
|------------|-------------|---------------------|-------------------------|
| MultiBLEU  | 36.5        | 21.5                | 36.8                    |

Table 6: Results of experiments on IWSLT 14 De-En about position-encoding. Word **PE** is short for position-encoding.

This evidence shows that here is no need to add position information to position-encoding because model views position-encoding as feature instead of converting them to order of sequence. It also shows that MoG cannot capture sequential information using position-encoding.

Position-encoding is still necessary for Graph-Transformer. Position-encoding is still the only component for Graph-Transformer to capture position information which is the same as the original Transformer. If one sentence contains one word multiple times, model can distinguish them at least. However, position-encoding cannot solve the problem that SAN-based model cannot capture localness information. **Position is not order.**

### A.8.2 WHY DOES PERFORMANCE OF MODEL DECREASE WHEN THE SENTENCE IS TOO LONG?

First, given a sentence, the MoG to reflect all relationship between words is definite and fixed, and cannot change. Performance of model reflects the ability to capture information for rebuilding this MoG from input sentence.

When the sentence is too long, models cannot capture enough information. As mentioned above, RNN-based model and SAN-based model cannot capture all subgraphs to rebuild the entire multigraph for different reason. Therefore, whether it is RNN-based model or SAN-based model, there should be a special length of sentence. If sentence is longer than this length, performance will drop very quickly.

Besides, in the training data, the proportion of long sentences is often very low, which makes model fail to learn long sentences.

### A.8.3 WHY PRE-TRAINING DATA CAN IMPROVE PERFORMANCE OF MODEL?

According to MoG explanation, word information cannot be enriched. In most of models, word information is generated by embeddings. A pre-training data feeds model better word information which can improve performance. Relationship information generated by using word information can also be more precise.

### A.8.4 WHAT IS NECESSARY TO DO DURING DESIGNING A NOVEL ENCODER?

A novel method to capture all subgraphs to rebuild the multigraph from the input sentence. Refer to RNN-based model and SAN-based model, and four basic questions in Section 2.5, we can summarize as follows.

- A proper architecture of model.
- A novel method should be proposed to calculate relationship between two given representations.
- This function should satisfy the modified law of distribution. Namely, given $r_a = \sum_i^n r_i^a$ and $r_b = \sum_j^m r_j^b$, function $f$ should satisfy the law of distribution

$$f(r_a, r_b) = \sum_i^n \sum_j^m w_{ij} f(r_i^a, r_j^b) \tag{11}$$

  where $w_{ij}$ is the weight for one relationship value.
- Several relationship can be added to one representation.
- $f$ does not satisfy the exchange law. Namely $f(a, b) \neq f(b, a)$.

- Several iterative processes should be proposed for generating representation. Generated representation should be feed to model for input representation and generate new representation.

### A.8.5   IS INFORMATION MODEL GENERATED ALL BENEFIT THE PERFORMANCE?

No. Some subgraph information like $a->b->a->b->a$ is one kind of noise which influences the performance of model.

