# OpenReview forum: "To Understand Representation of Layer-aware Sequence Encoders as Multi-order-graph"
_ICLR.cc/2021/Conference — Reject_

### Official Review · AnonReviewer3 · 2020-10-22
**Official Blind Review #3**

**Rating:** 6
**Confidence:** 4

**Review:**

Summary:

The paper proposes a new multigraph architecture called Multi-Order-Graph to explain the representation generation process in neural sequence encoders (Self-Attention or SAN based models). The main contribution of this MoG is the introduction of n-order dependency which can model not only relationships between words but also high order relationships such as syntax and semantics between subgraphs. Taking inspiration from MoG explanation, a self-attention powered Graph Transformer is proposed which beats the Transformer baselines on NMT tasks (English-German and German-English).

-----------------------------------------------------------------------------------------------------------------------------------------------------------------------------------
Pros:

+ The proposed idea of representing the encoding process as generation of a Multi-Order-Graph is novel and is able to provide good insights for SAN-based models.

+ Proposed Graph Transformer uses self-attention and also attends over different order of subgraphs (low-order, middle-order and high-order). Since these subgraphs represent high order relationships (syntax, semantics etc.), the model is able to pay attention to salient subgraphs.

+ This paper could fuel further research in the explainability of neural sequence encoders and SAN based models.

------------------------------------------------------------------------------------------------------------------------------------------------------------------------------------
Cons:

- The key concern about the paper is the lack of rigorous experimentation to study the usefulness of the proposed method. Only two datasets are used for NMT task. There has been a lot of improvement on base Transformer models (cited in the paper), still there is no comparison with them.

- This paper claims that with the proposed MoG explanation it is possible to model high order relationships such as Syntax and Semantics. However, there is no example provided for this. Even some cherry picked examples would have helped in visualising the behaviour.

------------------------------------------------------------------------------------------------------------------------------------------------------------------------------------

Overall, I would like to see the paper at the conference. The idea of modelling the relationships as a Multi-Order-Graph is certainly novel and can fuel further research. However, due to the current state of experimentation I am voting for **rejecting** the paper.  I am willing to increase the score if the concerns are addressed by authors in the rebuttal period.

------------------------------------------------------------------------------------------------------------------------------------------------------------------------------------

Questions and Suggestions:

1. I would like to see comparisons with models like Transformer(big), [Shaw et al.](https://arxiv.org/pdf/1803.02155.pdf), [He et al.](https://papers.nips.cc/paper/8019-layer-wise-coordination-between-encoder-and-decoder-for-neural-machine-translation.pdf). I would also encourage usage of some more datasets for the experimentation, for example En-Fr, En-Ro. Datasets can be found in the papers linked above.

2. It would be interesting to see how different components of the Graph Transformer affect perplexity.

3. What is Transformer(small)? I am sorry but I am not aware of this variation. It would be good if you could provide a reference for this particular variation of the model.

4. In Fig 4a, it is evident that self-gate works better at encoding longer sentences. It would be good to see some discussion on this in the paper.

5. Which dataset is used for Fig 4?

6. In fig 4b, we can notice a sudden decrease in subgraph weight at layer 4 for 12 layered model. In figure 5, similar trend can be seen for different layered model except for 9 and 10 layered model. What is the significance of this behaviour?

7. I would encourage the authors to provide the implementation for the model proposed in this paper.

------------------------------------------------------------------------------------------------------------------------------------------------------------------------------------

Minor Comments/Typos:
+ Section 2.4: source node of edge Snj -> Snj, source node of edge Ej
+ Section 2.4: target node of edge Tnj -> Tnj, target node of edge Ej
+ Section 3.3: the order of subgraph is in the range of 2^(n-1) to 2^(n) -> the order of subgraph is in range 2^(i-1) to 2^(i)
   Similar mistake is in Middle-Order and Low-Order section.
+ Section 7: MoG connects only only words but also subgraphs -> MoG connects not only words but also subgraphs.
+ A.5.3: Rlationship information generated -> Relationship information generated

Post Rebuttal Comments: Authors have addressed most of my concerns and as a result I have increased the rating from 5 to 6. Thanks!

---

> ### Author Response · Authors · 2020-11-19
> **Response to AnonReviewer3**
>
> Thank you for your review.
>
> ---
> ### About Example of Syntax
> Thank you for your valuable suggestions. We give an example about syntax. One syntactic tree can be viewed as one subgraph of MoG. Generating a syntactic tree can be viewed as a process of generating a subgraph.
>
> This example is updated to pdf in Figure 7.
>
>
> ---
> ### About experiments on other tasks and datasets
> To show effectiveness of the proposed MoG and Graph-Transformer, we have conducted several new experiments on NER, POS tagging, text summarization, and  additional machine translation language pairs: WMT14 English-French (En-Fr), and WMT16 English-Romanian (En-Ro). Please refer to General Response. Results of these experiments are updated to Table 1, Table 3 and Table 4 in our pdf.
>
>
> ---
> ### About Transformer (small)
> Transformer (small) is a specific set of hyperparameters which is designed for the low-resource IWSLT 14 De-En and shown in Table 3. To avoid confusion between it and Transformer (base), we call it Transformer (small).
>
> ---
> ### About self-gate
>
> Self-gate is based on self-attention which used in SAN-based model to capture edges and subgraphs. Whether previous or incremental representation is a set of subgraph information. In fact, weighting different representations by self-gate can be viewed as a process to capture edges between subgraphs in previous and incremental representation. It means that self-gate will generate subgraphs of higher order as we discussed in the last paragraph of Section 3.4, which can improve the performance on long sentence.
>
> ---
>
> ### About dataset in Figure 4
>
> WMT 14 En-De is the dataset used for Figure 4.
>
> ---
>
> ### About sudden decrease in subgrpah weight shown in Figure 5
>
> Weight reflects importance of representation. 4th-Layer in Graph-Transformer generates subgraphs of order in a range from 4 to 8 as incremental representation and subgraphs of order in a range from 1 to 4 as previous representation. This behaviour reflects that subgraphs of order in a range from 1 to 4 is more important. Considering weights of subgraphs generated in following layers, it means that 1~4-order dependency relationship is more important than other n-order dependency relationship.
>
>
> ---
> ### About Typos
> Thank you for your valuable suggestions. We will revise the next version according to your suggestions and correct these typos.

---

> ### Author Response · Authors · 2020-11-25
> **Response to AnonReviewer3-2**
>
> Thank you for your review.
>
> As for your concerned comparisons,
>
> Two extra language pairs are added in Table 1.
>
> PPL change has been added in Table 2.
>
> More baselines are from He and Shaw on Transformer-base are added in Table 1.
>
> We are sorry that with all we get, four nVida RTX GPUs, the experiment of Transformer-big on WMT 16 English-Romanian we conduct has not finished yet. So far, we observe the current training curves of Graph-Transformer and Transformer-big baseline are very close, which indicates that the improvement of Graph-Transformer on Transformer-big will not be so significant as that on Transformer-Base. However, such possible results are not surprising, and consistent with our MoG explanation.
>
> As we discussed in Section 2.4, the largest order of subgraph is limited by number of layers in SAN-based model. With 8 layers in Transformer-big, the largest order of generated subgraph is 256, which is larger than the length of longest sentence from training set and test set in WMT 16 En-Ro (240 and 140 respectively), which makes model tend to generate subgraph with the largest order. In the meantime, MoG shows that overlap of nodes and edges will happen as shown in Figure 1. Thus too many layers makes Graph-Transformer generate subgraphs with overlap, which may hinder the latent performance improvement.

---

> ### Author Response · Authors · 2020-11-25
> **Response to AnonReviewer3-3**
>
> Thank you for your review.
>
> The source code has been uploaded as Supplementary Material.

---

### Official Review · AnonReviewer4 · 2020-10-25
**idea is insightful, but some parts are not clear**

**Rating:** 6
**Confidence:** 3

**Review:**

The paper has propose the layer-aware graph transformer to enhance the ability of capturing heterogeneous information of the current models. The experimental results has justified the effectiveness of proposed model.

The multi-order graph structure is innovative and contribute to the expression of generating subgraphs. Such operations has benefited for questions the proposed in section 2.5, which I believe could bring up insightful idea to the community. However, there are still some parts that should be given an extended clear statement.

1, it is not clear how the "unified explanation" is defined and delivered in the graph-transformer model? Does it contribute the the model performance? MoG seems doesn't make a well-defined explainability to reflect the quantitative model explanation.

2, How MoG 'observe' every step of generation of embedding? Is there any model transparency defined?

3, Could the model be more generalized to other downstream task besides NMT?

4, It is not cleared that in the 2.2, whether it could be overlaps between SN and TN? If so, the number of generation of kinds of subgraph could be very large.

5, The experimental parts lack of detail procedure. How to extract the relations within each sentence? Also the performance of the proposed model may highly be dependent over the quality of MoG construction, which may limit the generalization ability of the model.

---

> ### Author Response · Authors · 2020-11-19
> **Response to AnonReviewer4**
>
> Thank you for your review.
>
> ---
> ### About unified explanation in graph-transformer
>
> In fact, MoG is the unified explanation of representation for layer-aware neural sequence encoders. Please refer to the contribution in our General Response.
>
>
> MoG generated by Graph-Transformer is similar as the MoG from Transformer while MoG from Graph-Transformer changes the weight of edge and enhances ability of capturing subgraph of high order.
>
> Four questions in Section 2.5 are good entries to analyze MoG and characteristics of one model. We can analyze Graph-Transformer in this way.
>
> 1. **How to preserve subgraph information?**   Graph-Transformer uses vector to preserve subgraph information same as other neural models.
> 2. **How to implement iterative encoding?**  Graph-Transformer uses Layer-level iteration only same as other SAN-based models, which means that (1) Graph-Transformer cannot capture order of sequence directly, (2) maximum order of subgraphs are limited in Graph-Transformer.
> 3. **How to capture edges and subgraphs?**  Graph-Transformer uses three self-attentions to capture edges and subgraphs which is the most fundamental difference between Graph-Transformer and the original Transformer.
> 4. **Which architecture will be selected?**  Architecture of Graph-Transformer is similar as the Transformer which has feed-forward layer, layer normalization and residual connection while Graph-Transformer have an additional component to combine representation generated by different self-attentions.
>
>
> With MoG, people can analyze, design and improve models more efficiently for the purpose of optimizing MoG and Graph-Transformer is an example of using MoG to improve the Transformer, which means that MoG explanation can be an effective tool to design and optimize layer-aware neural network model.
>
> An explanation of RNN-based model is updated to Section A.7 in pdf.
>
> ---
> ### About How MoG observer every steps of encoding
>
> Using MoG, we can make a detailed understanding of encoding, and simulate entire encoding process which can be regarded as a process of generating MoG. We can get some information of subgraphs such as when and where subgraph is generated. It helps use to understand the model and the representation more clearly.
>
> Although we can simulate the entire encoding process, it is difficult to observer every step of generation of embedding in detail. Number of subgraphs increases at an exponential rate and makes it expensive and difficult to observe the generating of each subgraph in a model. Besides, existed models always preserve multiple subgraph information in one vector which makes it difficult to distinguish,  extract observer one subgraph.
>
>
> ---
> ### About other downstream tasks
> Yes. We have conducted several new experiments on NER, POS tagging, text summarization, and  additional machine translation language pairs: WMT14 English-French (En-Fr), and WMT16 English-Romanian (En-Ro).  Please refer to General Response.
>
> Results of these experiments are updated to Table 1, Table 3 and Table 4 in our pdf.
>
> ---
> ### About overlaps between $SN$ and $TN$
> Yes, you are right. There can be overlaps between $SN$ and $TN$. This status is **Loop** which has been shown in Figure 1 (b). Loop is ubiquitous in MoG. For example, query vector and key vector can point to the same word in the Transformer which generates a loop.
>
>
> As we discussed in Section 2.4 and showed in Figure 1, the number of generation of kinds of subgraph is very large which makes it impossible to control all subgraphs.
>
>
> ---
> ### About some detail procedure
> Model can extract relationship between words within sentence during encoding, which is n-order dependency and actually building new edges in MoG. According to Section 3.1, this operation is implemented by self-attention in the original Transformer and Graph-Transformer.
>
>
> It is true that the performance of model may highly be dependent over the quality of MoG construction. However, from a perspective of MoG, an effective model should generate good MoG to reflect information captured from input as complete and accurate as possible, which means that quality of MoG construction is dependent over the ability of encoding of model basically.

---

### Official Review · AnonReviewer2 · 2020-10-27
**We carefully review the motivation, approach, and empirical results.**

**Rating:** 6
**Confidence:** 3

**Review:**

### Summary
The authors propose a new Transformer variant for neural machine translation. Compared with the standard Transformer framework, this work explains the representation generation process of the encoder via a multi-ordered-graph MoG and develops a novel Graph-Transformer method based on MoG, which is capable of capturing diverse relationships within the sequence. Empirical results over benchmark datasets validate the effectiveness of the proposed method.

### Pros
This work appears originally in its new explanation of the representation generation process. Specifically, the main pros are summarized below.
1.	It provides a multi-ordered-graph MoG explanation for the representation generation of Transformer encoder. MoG is capable of capturing diverse relations of the sequence compared with standard simple directly graph explanation.
2.	It develops a novel method, dubbed Graph-Transformer, by combining MoG and Transformer, which generates layer-wise representation from the previous representation and incremental representation aspects explicitly.
3.	It implements two fusion strategies, i.e., weight-gate and self-gate, for layer-wise information aggregation. Ablation study is also conducted to investigate the effectiveness of them.

### Cons
My primary questions/concerns are listed below.
1.	The experiments are limited with only two benchmark datasets. Since Transformer has been demonstrated to be effective in many NLP tasks, it would be better to include more datasets or even other tasks except neural machine translation for a comprehensive comparison.
2.	The selection of the baseline seems unfair. As stated in Section A.5.1, position-encoding can be viewed as a specific MoG because it builds edges between nodes. Therefore, methods that focus on position-encoding should also be included for comparison. For your reference, Transformer-XL[1] and [2] are two tailored solutions for this purpose.
3.	It is unclear about the model’s efficiency, i.e., memory-efficiency and learning efficiency. The idea to split the layer-wise full representation into previous representation and incremental representation is interesting. However, it also inevitably increases the model complexity. The authors should provide more discussions about the model complexity to improve the quality further.
4.	In Figure 3, why position-encoding is needed? MoG is already capable of capturing the sequential information by merging subgraphs. More explanation is welcomed for this configuration.

### Clarify
This work is well organized and easy to follow. However, the readability can be improved by addressing the following suggestions.
1.	The layer-level iteration in Figure 2 is difficult to understand; it would be better to give a toy example with a real sentence for a better explanation.
2.	In the second sentence of Section 2.4, r is not defined. In my understanding, each generated edge $e_j$ constructs a unique sub-graph, then what is r stand for?

[1] Dai, Zihang, et al. “Transformer-XL: Attentive language models beyond a fixed-length context”, ACL, 2019.
[2] Wang Benyou, et al. “Encoding word order in complex embeddings”, ICLR, 2020.


### Response to Rebuttal
Thank authors for taking the time to clarifications and considering my comments.

We appreciate authors' efforts to add additional experiments results in Table 1 and Table2. However, the performance improvements are marginal (more or less 0.7) and speed of Graph Transformer is slower than transformer.

Even though additional explanations about positional encoding (Appendix 8.1) can resolve our concerns, layer iteration (Figure 6) are not still clear for us, e.g., what are orange blue, yellow nodes? how layer iterations are used in graph transformer? Authors should refine its main context to increase understanding instead of adding lengthy Appendix for us. This such paper presentation and organization are not clear to understand.

Considering the above points, we still remain our decision.

---

> ### Author Response · Authors · 2020-11-19
> **Response to  AnonReviewer2**
>
> Thank you for your review.
>
> ---
> ### About Graph-Transformer on other NLP tasks and benchmark datasets
> To show effectiveness of the proposed MoG and Graph-Transformer, we have conducted several new experiments on NER, POS tagging, text summarization, and  additional machine translation language pairs: WMT14 English-French (En-Fr), and WMT16 English-Romanian (En-Ro).  Please refer to General Response-1.
>
> Results of these experiments are also updated to Table 1, Table 3 and Table 4 in our pdf.
>
>
>
> ---
> ### About position encoding in Graph-Transformer
> Position-encoding is still the only component for Graph-Transformer to capture position information and necessary. Graph-Transformer enhances ability of capturing subgraphs of high order while ability of capturing position information is same as the original Transformer.
>
>
> We have rewritten the Section A.8.1 to discuss position encoding from the perspective of MoG.
>
> ---
> ### About the efficiency of model
> Thank you for your reminding. We have recorded the size of model paragraphs and training speed and shown these data in Table 2 of pdf. It shows that it will cost slightly more time to train Graph-Transformer.
>
> ---
> ### About Layer-level iteration
> Left part of Figure 2 in paper is a brief schematic of layer-level iteration. Layer-level iteration takes representation generated by previous layer as input only, and new representation generated in one layer can only be used to generate subgraph in next layer which is different from sentence-level iteration. Please refer to General Response-1 for difference between layer-level iteration and sentence-level iteration.
>
> We update Figure 6 to explain the layer-level iteration in our pdf.
>
> Please refer to General Response for the difference between layer-level iteration and sentence-level iteration. This part is also updated to Section A.6 in our pdf.
>
>
> ---
> ### About $r$ in Section 2.4
> $r$ is short for **related subgraph**, and it is used in $sub^G_{j(r)}$ to represent the related subgraphs of edge $e_j$. To make it  clearer, we replace $sub^G_{j(r)}$ by $sub^G_{R(j)}$, where $R(j)$ is a function to get the identifier of related subgraph of $e_j$.

---

### Author Response · Authors · 2020-11-19
**General Response-1**

We thank all reviewers so much for the valuable comments on improving the quality of this work. We have updated the paper according to the review and our latest evaluations.


The revision primarily includes
1. We have conducted several new experiments on Named-entity recognition (NER), part-of-speech tagging (POS tagging), text summarization, and  additional machine translation language pairs.
2. We added the detailed descriptions about MoG and Graph-Transformer.
3. We added one new figure to describe the layer-level iteration.
4. We fixed some typo and writing problems.

---

### About Experiments
To show effectiveness of the proposed MoG and Graph-Transformer, we have conducted several new experiments on other tasks, Named-entity recognition (NER), part-of-speech tagging (POS tagging), text summarization, and  additional machine translation language pairs: WMT14 English-French (En-Fr), and WMT16 English-Romanian (En-Ro). We use CONLL2003 dataset, WSJ Corpus and Annotated Gigaword corpus as benchmark of NER task, POS tagging task and text summarization respectively. In these experiments, we use the original Transformer as the baseline.

|Model |  WMT14 En-Fr &nbsp;&nbsp;| WMT16 En-Ro|
|:----|:----:|:----:|
|Transformer (base)|40.1 |33.9|
|Graph-Transformer (halfdim-gate)&nbsp;&nbsp;|40.8 (+0.7)|34.6 (+0.7)|


|    Model     | NER (F1)|POS tagging (F1)|
|:----|:----:|:----:|
|Transformer |80.2 |96.40|
|Graph-Transformer (halfdim-gate)&nbsp;&nbsp;|80.3 (+0.1)|96.45 (+0.05)|

|Model|ROUGE-1|	ROUGE-2|	ROUGE-L|
|:----|:----:|:----:|:----:|
|Transformer (base)|	36.84	|18.01|	34.31|
|Graph-Transformer (halfdim-gate)&nbsp;&nbsp;|	37.38 (+0.54) |	18.59 (+0.58) |	34.58 (+0.27)|


This part will be updated to Table 1, Table 3 and Table 4 in our pdf.



---

### About Layer-level Interation and Sentence-level Interation

As we discussed in Section 2.4, process of subgraph (edge) generation is an iterative process, in which one subgraph (edge) relies on previous generated subgraphs (edges). Here is one question for this operation,

* **Where did previous generated subgraphs come from?**

which can also be transferred to another question,

* **Where will subgraphs generated be used to capture new subgraphs?**

Two kinds of iteration are answers to the question. Layer-level iteration means that one layer uses subgraph generated by previous layer to capture subgraph, and generated subgraph will be used in next layer. Sentence-level iteration means that one layer uses subgraphs generated in the same layer at previous time step, and generated subgraph will be used in the same layer at next time step.

Note that layer-level iteration is not the opposite of sentence-level iteration. They can be used together in one model, or can exist independently in one model. However, in most of layer-aware model with multiple layers, layer-level iteration is necessary.

This part is updated to pdf in Section A.6


---

### About Contributions

Finally, we want to clarify our contributions.

Our main contribution in this work is providing a unified explanation of representation layer-aware neural sequence encoder, which regards representation as a revisited multigraph called multi-order-graph (MoG), so that model encoding can be viewed as a processing to capture all subgraphs in MoG. MoG can explain not only  SAN-based model, but also other layer-aware models such as RNN-based model.


Specially, we propose Graph-Transformer based on MoG explanation. It is designed to show an example of application of MoG explanation which is the main purpose of Graph-Transformer. Besides, it can also improve the performance of the Transformer and results of experiments on several NLP tasks show the effectiveness of Graph-Transformer.

---

### Comment · Area_Chair1 · 2020-11-22
**Time for discussion**

Dear Reviewers,

The authors have provided a detailed response and updated their revised manuscript. Would you please take a careful look at their response and update your review accordingly?

Thanks,
AC

---

### Author Response · Authors · 2020-11-25
**Revision Summary**

We thank the reviewers for their time and valuable suggestion. Reviews noted that we propose Multi-Order-Graph to explain the representation generation process, and propose   Graph-Transformer for neural machine translation, and evaluate Graph-Transformer in WMT 14 En-De and IWSLT 14 De-En. However, the reviewers also have some curiosities about explanation in our paper, and concern about our experiments on other tasks and benchmark datasets. To address these concerns, based on the helpful suggestions and constructive questions of the reviewers, we made substantial changes to the paper organization, MoG explanation, useful examples, baseline model, and comparisons to related works, and experiments on Named-entity recognition, part-of-speech tagging, text summarization and machine translation language pairs WMT 14 En-Fr and WMT 16 En-Ro to show the effectiveness of proposed Graph-Transformer. We have uploaded a new version in which more clarification and more experiments reviewers suggested were added.

The modification can be briefly summarized as follows:

1. Add one section to explain the difference between layer-level iteration and sentence-level iteration (see Appendix Section A.6), and one figure to show the layer-level iteration (see Appendix Figure 6)
2. Modify and enrich the explanation of position-encoding in the Transformer. (see Appendix Section A.8.1).
3. Add one section to explain RNN-based model using MoG explanation. (see Appendix  Section A.7).
4. Add one figure to show an example of modelling syntax as $n$-dependency relationship. (see Appendix Figure 7).
5. Add results of experiments on Named-entity recognition (see Section 4.2 and Table 3) and part-of-speech tagging (see Section 4.2 and Table 4).
6. Add results of experiments on two benchmark datasets, WMT 14 En-Fr and WMT 16 En-Ro (see Section 4.1 and Table 1).
7. Add the comparison of the effects related works, Shaw et al. and He et al. (see Section 4.1 and Table 1).
8. Add data of the size of model paragraphs, training speed and PPL (see Section 4.1 and Table 2).
9. Add discussion about Transformer-big in the comments *Response to AnonReviewer3-2*.

---

### Decision · Program_Chairs · 2021-01-07
**Final Decision**

**Decision:**

Reject

**Comment:**

The paper proposes to explain the representation for layer-aware neural sequence encoders with multi-order-graph (MoG). Based on the MoG explanation, it further proposes Graph-Transformer as a graph-based self-attention network empowered Transformer. As commented by the authors, a main purpose of Graph-Transformer is to show an example application of the MoG explanation.

During the discussion period, after reading the paper and checking the code, the AC had raised a serious concern: There is a big gap between the MoG motivation and the actual implementation. The AC had urged the referees to take a careful look at the implementation details, in particular, Lines 524-561 in the attached code: "supplement/fairseq-0.6.2_halfdim_gate⁩ ▸ ⁨fairseq⁩ ▸ ⁨models⁩ ▸transformer.py". The AC had made the following comments to the referees: "Whether the performance gain of Graph-Transformer over Transformer is due to the MoG explanation is highly unclear. There is no direct evidence, such as appropriate visualization, to support that. In a high-level description, instead of using a usual skip connection that would combine beforex and x, the actual implementation is to 1) define increamental_x = x - beforex, 2) let increamental_x attend on beforex to produce x1, let beforex attend on increamental_x to produce x2, and let increamental_x attend on increamental_x to produce x3, 3) combine beforex, x1, x2, x3 in a certain way to produce the layer output."

Reviewer 2 responded to the AC's concern: "After examining the transformer.py and Section 2 & 3, we cannot understand why the output of self-attentions could be regarded as MoG subgraphs? The authors did not explain the connection. In their code, the graph transformer seems to just utilize 3 multi-head attentions (line 539-541) in their encoder. Using MoG to interpret the outputs of three attentions (line 539-541) is not very convincing. The link is weak. We agree with your comments."

To summarize, the link between the actual implementation in the code and all the MoG explanations is quite weak, and the technical novelty of the actual implementation is not strong enough for an ICLR publication. Therefore, the AC recommends Reject.